

# On the stability interpretation of Extended Column Test results

Frank Techel[1,2], Kurt Winkler[1], Matthias Walcher[1,3], Alec van Herwijnen[1], and Jürg Schweizer[1]

[1]WSL Institute for Snow and Avalanche Research SLF, Davos, Switzerland
[2]University of Zurich, Department of Geography, Zurich, Switzerland
[3]currently independent researcher

**Correspondence:** Frank Techel (techel@slf.ch)

**Abstract.** Snow instability tests provide valuable information regarding the stability of the snowpack. Test results are key data used to prepare public avalanche forecasts. However, to include them into the operational procedures, a quantitative interpretation scheme is needed. Whereas the interpretation of the Rutschblock test is well established, a similar detailed classification for the Extended Column Test (ECT) is lacking. Therefore, we develop a 4-class stability interpretation scheme.

Exploring a large data set of 1719 ECTs observed at 1226 sites, often performed together with a Rutschblock (RB) in the same snow pit, and corresponding slope stability information, we revisit the existing stability interpretations, explore the potential of a more detailed classification, and specifically consider the interpretation of cases when two ECTs were performed in the same snow pit. Our findings confirm previous research, namely that the crack propagation propensity is the most relevant result and that the loading step required to initiate a crack is of secondary importance for stability assessment. The comparison with

the RB showed that the ECT classifies slope stability less reliably than the RB. In some situations, performing a second ECT may be helpful, when the first test did neither indicate rather unstable nor stable conditions. Finally, the data clearly show that false-unstable predictions of stability tests outnumber the correct-unstable predictions in an environment where overall unstable locations are rare.

*Copyright statement.* TEXT

## 1 Introduction

Gathering information about current snow instability is crucial when evaluating the avalanche situation. However, direct evidence of instability - as recent avalanches, shooting cracks or whumpf sounds - is often lacking. When such clear indications of instability are absent, snow instability tests are widely used to obtain information on the stability of the snowpack. Such tests provide information on failure initiation and subsequent crack propagation - essential components for slab avalanche release

(Schweizer et al., 2008b; van Herwijnen and Jamieson, 2007). However, performing snow instability tests is time-consuming, as they require to dig a snow pit. Furthermore, considerable experience in the selection of a representative site is needed, and the interpretation of test results is challenging (Schweizer and Jamieson, 2010). Alternative approaches such as interpreting snow micro-penetrometer signals (Reuter et al., 2015; van Herwijnen et al., 2009), are promising, but not sufficiently estab-



lished yet.

Two commonly used tests to assess snow instability are the Rutschblock test (RB, Föhn, 1987) and the Extended Column Test (ECT; Simenhois and Birkeland, 2006, 2009). For both tests, which are described in greater detail in Section 2.1, blocks of snow are isolated from the surrounding snowpack. According to test specifications, the block is then loaded in several steps. The loading step leading to a crack in a weak layer (failure initiation) is recorded, and whether crack propagation across the entire block of snow occurs (crack propagation). For the RB, the interpretation of the test result is well established and involves

combining failure initiation (score) and crack propagation (release type) (e.g. Schweizer, 2002; Winkler and Schweizer, 2009). In contrast, the original interpretation of ECT results considers crack propagation propensity only (Simenhois and Birkeland, 2006, 2009; Ross and Jamieson, 2008): if a loading step leads to a crack propagating across the entire column, the result is considered as *unstable*, else as *stable*. However, Winkler and Schweizer (2009) suggested to improve this binary classification by additionally considering the loading step required to initiate a crack and by considering a minimal failure layer depth leading

to interpretations of ECT results as *unstable*, *intermediate* and *stable*. Moreover, they hypothesized that performing two tests, and considering differences in test results, may help to establish an intermediate stability class.

As the properties of the slab as well as the weak layer may vary on a slope (Schweizer et al., 2008a), reliably estimating slope stability requires many samples (Reuter et al., 2016) and a single test result may not be indicative. Hence, it was suggested to perform more than one test, either in the same snow pit or in a distance beyond the correlation length, which is often on

the order of ≤ 10 m (Kronholm et al., 2004). For instance, Schweizer and Bellaire (2010) analysed whether performing two pairs of Compression Tests (CT) about 10 m apart improves slope stability evaluation. They suggested a sampling strategy that essentially suggests that in case the first test does not indicate instability, additional tests can reduce the number of false-stable predictions. Moreover, they reported that in 61–75% of the cases the two tests in the same pit provided consistent results, in the remaining cases either the CT score or the fracture type varied. For the ECT, several authors also noted that two tests performed

adjacent to each other in the same snow pit or at several meters distance within the same small slope frequently lead to different results (Winkler and Schweizer, 2009; Hendrikx et al., 2009; Techel et al., 2016). For instance, Techel et al. (2016) reported that in 21% of the cases the ECT fracture propagation result differed between two tests in the same snow pit. Moreover, they explored differences in the performance between the ECT and the RB with regard to slope stability evaluation and found that the RB detected more stable and unstable slopes correctly than a single ECT or two adjacent ECTs.

Both, ECT and RB provide information relating to slab avalanche release. While the Rutschblock provides reliable results, the ECT is quicker to perform in the field, which probably explains why it has quickly become the most widely used instability test in North America (Birkeland and Chabot, 2012). Given the popularity of the ECT as a test to obtain snow instability information and the lack of a quantitative interpretation scheme that includes more than just two classes, our objective is to revisit the originally suggested stability interpretations and to specifically consider cases when two ECTs were performed in

the same snow pit. Building on our findings, we propose a new stability classification differentiating between cases when just a single ECT and when two adjacent ECTs were performed in the same snow pit with the goal to minimize false-stable and false-unstable predictions. Additionally, we empirically explore the influence of the base rate, the frequency of unstable locations, on stability test interpretation, which - if neglected - may lead to false interpretations (Ebert, 2018). We address this





**Table 1.** Data overview with the number (N) and proportion of *unstable* rated slopes.

| stability tests | N | *unstable* |
|---|---|---|
| single ECT | 279 | 15% |
| two ECT | 208 | 30% |
| single ECT and a RB | 454 | 20% |
| two ECT and a RB | 285 | 20% |

topic by exploring a large set of ECT with observations of slope stability collected in Switzerland. Furthermore, ECT results

are compared with concurrent RB test results.

## 2 Data

Data were collected in 13 winters from 2006-2007 to 2018-2019 in the Swiss Alps. We explored a data set of stability test results (Sect. 2.1 and 2.2) in combination with information on slope stability (Sect. 2.3) and avalanche hazard (Sect. 2.4). At 1226 sites, for which slope stability information was available, 1719 ECT were performed (Tab. 1). At 487 out of the 1226

sites either one (279) or two ECTs (208) were performed (695 ECTs in total). At the other 739 sites, a RB test was conducted in addition to either one (484) or two ECTs (285) in the same snow pit (1024 ECTs in total).

### 2.1 Extended Column Test (ECT) and Rutschblock test (RB)

At sites where ECT and RB were realized in the same snow pit, one or two ECTs were generally performed directly down-slope from the RB (e.g. as described in detail in Winkler and Schweizer (2009)). If no RB was performed but two ECTs were

performed, it is not known whether the ECTs were performed side-by-side, or whether the second ECT was located directly up-slope from the first ECT.

Test procedure followed observational guidelines (Greene et al., 2016). For the ECT, loading is by tapping on the shovel blade positioned on the snow surface on one side of the column of snow isolated from the surrounding snowpack (30 loading steps, Fig. 1a). For the RB, a person on skis stands or jumps on the block (6 loading steps, Fig. 1b). When a crack initiates and

propagates within the same weak layer across the entire column within one tap of crack initiation, it is called *ECTP* for the ECT; for the RB this corresponds to the release type *whole block*. If the crack does not propagate within the same layer across the entire column or within one tap of crack initiation, *ECTN* is recorded for the ECT. Similarly, if the fracture does not propagate through the entire block, *part of block* or *edge only* are recorded as RB release type. If no failure can be initiated including loading step 30 (ECT) or 6 (RB), these are recorded as *ECTX* or *RB7*, respectively.




## 2.2 Stability classification of ECT and RB

To facilitate the distinction between the result of an instability test and the stability of a slope, we refer to test stability using four classes 1 to 4, with class 1 being the lowest stability (*poor* or less) and class 4 the highest stability (*good* or better). In contrast, for slope stability, we use the terms *unstable* and *stable*. We chose four classes as a similar number of classes has been used for RB stability interpretation, as outlined below.

**Extended Column Test (ECT):** The stability classification originally introduced by Simenhois and Birkeland (2009) ($ECT_{orig}$) suggested two stability classes: *ECTN* or *ECTX* are considered to indicate high stability (class 4), while *ECTP* indicates low stability (class 1).

The classification suggested by Winkler and Schweizer (2009) ($ECT_{w09}$) uses three classes:

- *ECTP*≤21: low stability (class 1)

- *ECTP*>21: intermediate stability (class 2-3)

- *ECTN* or *ECTX*: high stability (class 4)

**Rutschblock test:** We classified the RB in four classes (classes 1 to 4; Fig. 2). We followed largely the RB stability classification by Techel and Pielmeier (2014), who used a simplified version of the classification used operationally by the Swiss avalanche warning service (Schweizer and Wiesinger, 2001; Schweizer, 2007). Schweizer (2007) defined five stability classes for the RB, based on the score and the release type in combination with snowpack structure, while Techel and Pielmeier (2014) relied exclusively on RB score and release type. In contrast to both these approaches, we combined the two highest classes (*good* or *very good*) to one class (class 4).

Shallow weak layers (≤ 15 cm) are rarely associated with skier-triggered avalanches (Schweizer and Lütschg, 2001; van Herwijnen and Jamieson, 2007), which is, for instance, reflected in the threshold sum approach (Schweizer and Jamieson, 2007), a method to detect structural weaknesses in the snowpack. Schweizer and Jamieson (2007) reported the critical range for weak layers particularly susceptible to human triggering as 18-94 cm below the snow surface. Minimal depth criteria were also taken into account by Winkler and Schweizer (2009) in their comparison of different instability tests or by Techel and Pielmeier (2014), when classifying snow profiles according to snowpack structure. We addressed this, by assigning the next higher stability class if the weak layer was between 6 and 10 cm below the surface, and class 4 if the failure layer was less than 5 cm below the snow surface. If there were several failure planes in the same test, we searched for the ECT and RB failure plane with the lowest stability class.

## 2.3 Slope stability classification

We classified stability tests according to observations relating to snow instability in similar slopes as the test on the day of observation, such as recent avalanche activity or signs of instability (whumpfs or shooting cracks). This information was manually extracted from the text accompanying a snow profile and/or stability test. This text contains - among other information -



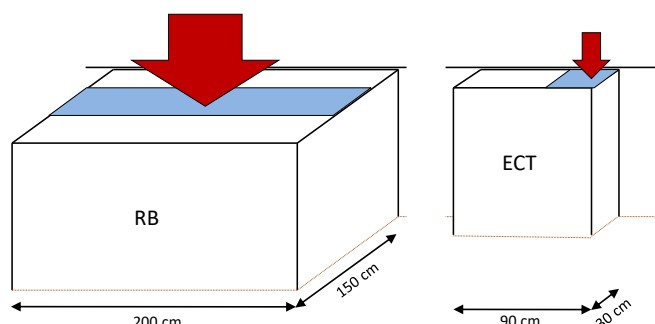

**Figure 1.** ECT and RB according to observational guidelines. At the back, the block of snow is isolated by either cutting with a cord or a snow saw. The lightblue area indicates the approximate area, where the skis or the shovel blade is placed. This area corresponds to the area loaded for the ECT, while the main load under the skis is exerted over a length of about 1 m (Schweizer and Camponovo, 2001). Loading is from above (arrows).

| RB | | score | | | | | | |
|---|---|---|---|---|---|---|---|---|
| | | 1 | 2 | 3 | 4 | 5 | 6 | 7 |
| release type | whole block | 1 | | 2 | 3 | | | |
| | partial release* | | 2 | 3 | 4 | | | |

**Figure 2.** Classification of RB into four stability classes. *combines release type *part of block* and *edge only*.

details regarding recent avalanche activity or signs of instability.

A slope was considered *unstable* if any signs of instability or recent avalanche activity - natural or skier-triggered avalanches from the day of observation or the previous day - were noted on the slope where the test was carried out or on neighbouring

slopes (Simenhois and Birkeland, 2006, 2009; Moner et al., 2008; Winkler and Schweizer, 2009; Techel et al., 2016).

We considered a slope only as *stable*, if it was clearly stated that on the day of observation none of the before-mentioned signs were observed in the surroundings. In most cases, surroundings relates to observations made in the terrain covered or observed during a day of back-country touring (estimated to be approximately 10 to 25 km$^2$, Meister, 1995; Jamieson et al., 2008).

In the following, we denote slope stability simply as *stable* or *unstable*, although this strict binary classification is not entirely

correct. For instance, many tests were performed on slopes that were actually rated as *unstable*, though did not fail.

If it was not clearly indicated, when and where signs of instabilities or fresh avalanches were observed, or if this information was lacking entirely, these data had not been included in our dataset.

## 2.4 Forecast avalanche danger level

For each day and location of the snow instability test, we extracted the forecast avalanche danger level related to dry-snow conditions from the public bulletin issued at 17.00 CET, and valid for the following 24 hours.





## 3 Methods

### 3.1 Criteria to define ECT stability classes

We consider the following criteria as relevant when testing existing or defining new ECT stability classes:

– (*i*) Stability classes should be distinctly different from each other. The criteria we rely on is the proportion of *unstable* slopes. Therefore, a higher stability class should have a significantly lower proportion of *unstable* slopes than the neighboring lower stability class.

    – (*ii*) The lowest and highest stability classes should be defined such that the rate of correctly detecting *unstable* and *stable* conditions is high, respectively; hence, the rate of *false-stable* and *false-unstable* predictions should be low, respectively.

Stability classes in-between these two classes may represent *intermediate* conditions, or lean towards more frequently *unstable* and *stable* conditions, permitting a higher *false-stable* and *false-unstable* rate than the rates of the two extreme stability classes.

    – (*iii*) The extreme classes should occur as often as possible, as the test should discriminate well between *stable* and *unstable* conditions in most cases.

To define classes based on crack propagation propensity and crack initiation (number of taps), we proceeded as follows:

1. We calculated the mean proportion of *unstable* slopes for moving windows of 3, 5 and 7 consecutive number of taps, for ECTP and ECTN separately. ECTX was included in ECTN, treating ECTX as ECTN31.

2. We obtained thresholds for class intervals by applying unsupervised kmeans-clustering (R-function *kmeans* with settings max.iter = 100, nstart = 100; R Core Team (2017); Hastie et al. (2009)) on the proportion of *unstable* slopes of the three

running means (step 1). The number of clusters *k* tested were 3, 4 and 5.

3. We repeated clustering 100 times using 90% of the data, which were randomly selected without replacement. For each of these repetitions, the cluster boundaries were noted. Based on the 100 repetitions, we report the respective most frequently observed *k*-1 boundaries, together with the second most frequent boundary.

4. To verify whether the classes found by the clustering algorithm were distinctly different (criteria *i*), we compared the

proportion of *unstable* slopes between clusters using a two-proportions z-test (*prop.test*, R Core Team (2017)). We considered p-values $\leq 0.05$ as significant.

    In almost all cases, we used a one-sided test with the null hypothesis $H_0$ being either $H_0$: $prop$(A) $\leq$ $prop$(B) (or its inverse), where $prop$ is the proportion *unstable* slopes in the respective cluster A or B. The alternative hypothesis $H_a$ would then be $H_a$: $prop$(A) $>$ $prop$(B) (or its inverse).

5. For clusters not leading to a significant reduction in the proportion of *unstable* slopes, we tested a range of thresholds ($\pm$ 3 taps within the threshold indicated by the clustering algorithm) to find a threshold maximizing the difference between





cluster centers and leading to significant differences ($p \leq 0.05$) in the proportion of *unstable* slopes (criteria *ii*). If no such threshold could be found, clusters were merged.

Throughout this manuscript, we report p-values in four classes ($p > 0.05$, $p \leq 0.05$ when p = [0.05,0.01[ , $p \leq 0.01$ when p = [0.01,0.001[ and $p \leq 0.001$).

### 3.2 Assessing the performance of stability tests and their classification

When the predictive power or predictive validity of a test is assessed, it is compared to a reference standard, here the slope stability classified as either *unstable* or *stable*. The usefulness of instability test results is generally assessed by considering only two categories related to *unstable* and *stable* conditions (Schweizer and Jamieson, 2010). We refer to these two outcomes as *low* or *high* stability.

There are two different contexts a test's adequacy is looked at: the first explores whether the foundations of a test are satisfactory (i), the second its practical usefulness (ii) (Trevethan, 2017):

(i) Most often the performance of a snow stability test is assessed from the perspective of the reference group (Schweizer and Jamieson, 2010), i.e. what proportion of *unstable* slopes are detected by the stability test. The two relevant measures addressing this context are the sensitivity and specificity, which are considered as the benchmark for the performance:

– The sensitivity of a test is the probability of correctly identifying an *unstable* slope from the slopes that are known to be *unstable*. Considering a frequency table (Tab. 2) the sensitivity, or probability of detection (POD), is calculated as (Trevethan, 2017):

$$\text{Sensitivity (POD)} = \frac{a}{a+c}$$

– The specificity of a test is the probability of correctly identifying a *stable* slope from the slopes that are known to be *stable*. It is also referred to as probability of non-detection (PON).

$$\text{Specificity (PON)} = \frac{d}{b+d}$$

Ideally, both sensitivity and specificity are high, which means that most *unstable* and most *stable* slopes are detected. However, missing *unstable* situations can have more severe consequences and therefore it is assumed that first of all the sensitivity should be high. Nonetheless, a comparably low specificity will decrease a test's credibility. Sensitivity and specificity are generally considered to be insensitive to the distribution of reference standard - in our case the respective proportions of *unstable* and *stable* slopes. However, this is only true when the distribution of the reference classes is approximately balanced and misclassifications in the estimated reference classes are rare (Brenner and Gefeller, 1997).

(ii) The second context focuses on the ability of a test to correctly indicate slope stability, i.e. if the test result indicates low stability, how often is the slope in fact *unstable*. This aspect has only rarely been explored for snow instability tests (e.g by Ebert (2018) from a Bayesian viewpoint), and is generally assessed using two metrics:





– The positive predictive value (PPV) is the proportion of *unstable* slopes, given that a test result indicates instability (a low stability class).

$$\text{PPV} = \frac{a}{a+b}$$
$$= \text{proportion } \textit{unstable} \text{ slopes}$$

is the statistic we refer to most in this manuscript, generally termed the proportion of *unstable* slopes.

– The negative predictive value (NPV) is the proportion of *stable* slopes, given that a test result indicates stability (a high stability class).

$$\text{NPV} = \frac{d}{c+d}$$
$$= \text{proportion } \textit{stable} \text{ slopes}$$

PPV and NPV are correlated to the distribution of *unstable* and *stable* slopes in the data set. Thus keeping the base rate the same when making comparisons across tests and stability classifications is essential.

However, to demonstrate the effect of a varying base rate, we highlight differences in PPV and NPV by considering the proportion of *unstable* slopes stratified by the forecast danger level for 1-Low to 3-Considerable.

Finally, a test result should provide interpretable evidence in favour of instability or stability. To address this point, we use the

likelihood ratio as a measure of the strength of evidence for one hypothesis or the other. According to Brenner and Gefeller (1997), and applied to our study, the positive likelihood ratio LR+ is the ratio of the probability of a positive test (*low* stability) in an *unstable* slope to the probability of a positive test in a *stable* slope:

$$\text{LR+} = \frac{a/(a+c)}{b/(b+d)}$$
$$= \frac{\text{POD}}{1-\text{PON}}$$

The likelihood ratio is the factor that describes the shift from the prior probabilities to the posterior probabilities, and is therefore an indicator of the strength of evidence the observed data have (Blume, 2002).

### 3.3 Base rate of *unstable* and *stable* slopes

As outlined before, the proportion of *unstable* slopes varied within our data set: We noted a bias towards more frequently observing two ECTs when the slope stability was considered as *unstable* (30%), compared to single ECT with only 15% of the

tests observed in *unstable* slopes (Table 1). To balance out this mismatch when comparing two ECT results to single ECT or RB (20% *unstable*), we created equivalent data sets for single ECT and RB containing the same proportion of tests collected on *unstable* and *stable* slopes as the data set of two ECT. For this, we randomly sampled an appropriate number of single




**Table 2.** 2×2 frequency table cross-tabulating slope stability and test results. A positive test result indicates *low* stability, a negative test result *high* stability.

|  |  | slope stability | |
|---|---|---|---|
|  |  | *unstable* | *stable* |
| test result (*stability*) | positive (*low*) | a | b |
|  | negative (*high*) | c | d |

ECT and RB observed on *stable* slopes, and combined these with all the tests observed on *unstable* slopes. We repeated this procedure 100 times. We report only the mean values of these 100 repetitions. P-values (prop.test) were calculated for these
mean proportions and the original number of cases in the data set.

The base rate with 30% tests on *unstable* and 70% on *stable* slopes was used throughout this manuscript, except in Sect. 4.5, where we evaluate the effect of different base rates.

### 3.4 Selecting ECT from snow pits with two ECT

For snow pits with two adjacent ECTs, we randomly selected one ECT, when exploring single ECT data or the relationship
between the number of taps and slope stability (Sect. 4.2). As before, this procedure was repeated 100 times. The respective statistics, generally the mean proportion of *unstable* slopes, was calculated based on the 100 repetitions.

## 4 Results

### 4.1 Comparing existing stability classifications

We first consider the results for a single ECT.
The original stability classification $ECT_{orig}$ led to significantly different proportions of *unstable* slopes for the two stability classes (0.47 vs. 0.18, p < 0.001, Fig. 3a). The $ECT_{w09}$-classification, with three different classes, showed significantly different proportions of *unstable* slopes between the lowest and the intermediate classes (0.53 vs. 0.23, p ≤ 0.001), but not between the intermediate and the highest classes (0.23 and 0.18, p > 0.05). Although $ECT_{w09}$-class 1 had a larger proportion *unstable* slopes than $ECT_{orig}$-class 1, the difference was not significant (p > 0.05).
Considering the results obtained from two adjacent ECTs resulting in the same stability class 1, between 0.52 ($ECT_{orig}$) and 0.61 ($ECT_{w09}$) of the slopes were *unstable*. Although the proportion of *unstable* slopes was higher by 0.05 to 0.08 than for a single ECT, this difference was not significant (p > 0.05). When both ECT indicated the highest stability class, the proportion of *unstable* slopes was 0.15, not significantly different than for a single ECT resulting in this stability class (0.18, p > 0.05). When one test resulted in the lowest and the other in the intermediate $ECT_{w09}$-class, 0.25 of the slopes were *unstable*. While
this was clearly less than when both resulted in $ECT_{w09}$-class 1 (p < 0.05), it was not significantly different than two ECT with





*ECT$_{w09}$*-class 4 (0.15, p > 0.05)

Regardless whether a single ECT or two ECTs were considered, the *ECT$_{w09}$*-classification had a 0.06-0.09 larger proportion of *unstable* slopes for stability class 1 than the *ECT$_{orig}$*-classification. For stability class 4 there was no difference, as the definition for this class was identical.

The sensitivity was higher for *ECT$_{orig}$* (0.64) than for *ECT$_{w09}$* (class 1: 0.57, Fig. 4a and b). However, this comes at the cost of a high false alarm rate (1-specificity) for *ECT$_{orig}$* (0.31), considerably higher than for *ECT$_{w09}$* (0.21).

The optimal balance between achieving a high sensitivity and a low false alarm rate was found to be at ECTP≤21 (R-library *pROC* (Robin et al., 2011)), exactly the threshold suggested by Winkler and Schweizer (2009).

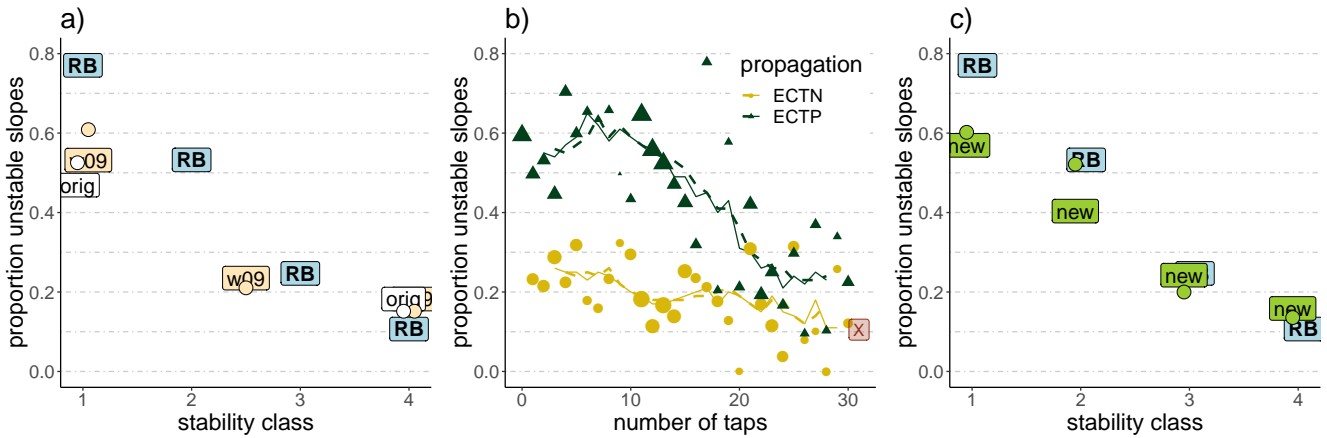

**Figure 3.** Proportion of *unstable* slopes (y-axes) for a) the two existing ECT stability classifications (*ECT$_{orig}$*, *ECT$_{w09}$*) and the RB, b) the number of taps stratified by propagation, and c) the classification using the *ECT$_{new}$* together with the RB as in a). In a) and c): single ECT are indicated by the respective text labels, two ECTs resulting in the same stability class by points. In b): The lines represent the mean proportion of *unstable* slopes calculated for moving windows including five or seven consecutive number of taps. a) to c) 30% unstable and 70% stable slopes were used.

## 4.2 Clustering ECT results by accounting for failure initiation and crack propagation

So far, we explored existing classifications. Now, we focus on the respective lowest number of taps stratified by propagating (*ECTP*) and non-propagating (*ECTN*) results. If in the same test for different weak layers ECTN and ECTP were observed, only ECTP with the lowest number of taps was considered.

As can be seen in Fig. 3b, the proportion of *unstable* slopes was higher for *ECTP* compared to *ECTN*, regardless of the number of taps and in line with the original stability classification *ECT$_{orig}$*. However, a notable drop in the proportion of *unstable* slopes

between about 10 and 25 taps is obvious (ECTP, from about 0.6 to almost 0.25).

Clustering the ECT results shown in Figure 3b with the number of clusters *k* set to 3, 4 and 5, and repeating the clustering 100 times, each time with 90% of the data, split the data at similar thresholds. In the following, we show the results for the

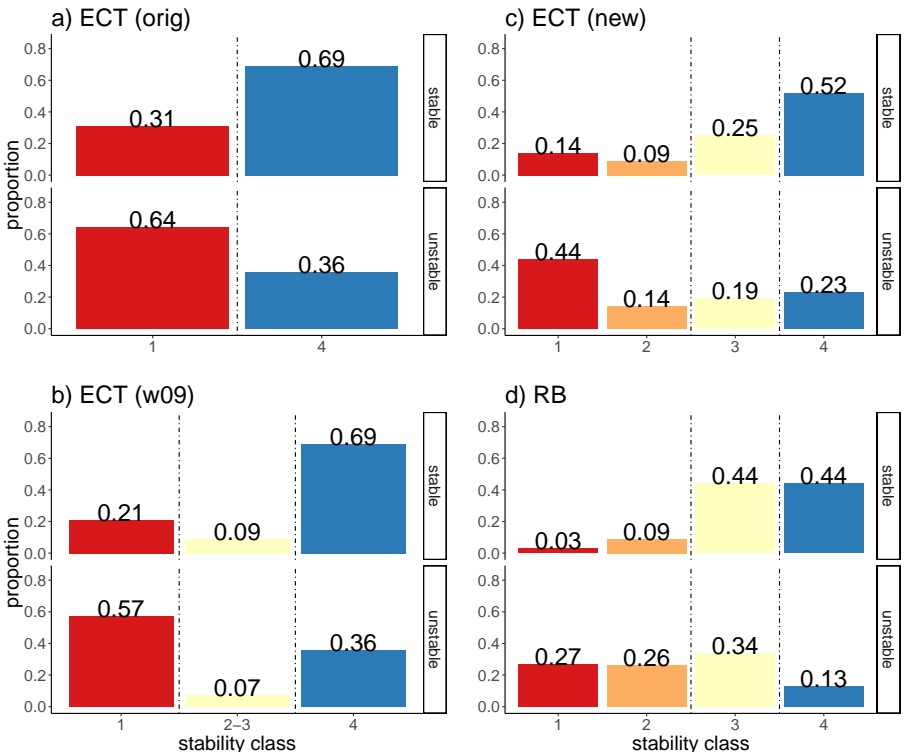

**Figure 4.** Distribution of stability classes by slope stability for the different stability test and classification approaches: a) with two classes (ECT$_{orig}$); b) with three classes (ECT$_{w09}$); and c) and d) with four classes (ECT$_{new}$, Rutschblock, respectively). The vertical dashed lines indicate the thresholds when the primary slope stability associated with a test result changed from one slope stability to the other. - Reading subfigures row-wise provides an indication of POD and PON. Comparing proportions column-wise corresponds to a base rate of 0.5. If no clear prevalence (values between 40 and 60%) was observed, the stability class is considered as intermediate (light yellow colour).

two most frequent cluster thresholds obtained for $k = 4$. The frequency, the respective cluster threshold was selected in the 100 repetitions, is shown in brackets:

– ECTP≤15 (48%), ECTP≤14 (36%)

     – ECTP≤20 (37%), ECTP≤18 (36%)

     – ECTN≤10 (29%), ECTN≤9 (22%)

Setting $k$ to 3 resulted in clusters being divided at ECTP≤14 and at ECTP≤21, $k = 5$ resulted in cluster thresholds ECTP≤9, ECTP≤14, ECTP≤20 and ECTN≤10. The second most frequent threshold was almost always within ±1 tap of those indicated 265   before.

To maximize the difference in the proportion of *unstable* slopes between classes (Fig. 3c), we varied the thresholds defining


clusters by testing ±3 taps. The following four stability classes for single ECT ($ECT_{new}$) were obtained (p-values indicate whether the proportion of *unstable* slopes differed in relation to the previously described group):

1. ECTP≤13 - capturing test results with the largest proportion of *unstable* slopes. The proportion of *unstable* slopes (0.57) was about double the base rate (0.3).

2. ECTP>13 and ECTP≤22 (proportion of *unstable* slopes = 0.4, p ≤ 0.05) - transitioning from a high (0.57, for ECTP≤13) to a lower proportion of *unstable* slopes (0.23, for ECTP>22). However, the mean proportion of *unstable* slopes was still higher than the base rate.

3. ECTP>22 or ECTN≤10 (0.23, p ≤ 0.01) - the proportion of *unstable* slopes was lower than the base rate.

4. ECTN>10 or ECTX (0.15, p ≤ 0.05) - capturing test results corresponding to the lowest proportions of *unstable* slopes (about half the base rate).

In the following, we apply these thresholds in combination with the depth of the failure plane.

### 4.3 Evaluating the new ECT stability classification

#### 4.3.1 Stability classification for single ECT

The new classification with four stability classes ($ECT_{new}$) showed continually and significantly decreasing proportions of *unstable* slopes with increasing stability class (0.57, 0.39, 0.25, 0.16 for classes 1 to 4, respectively, p ≤ 0.01, Fig. 3c).
The lowest $ECT_{new}$-class had a larger proportion *unstable* slopes (0.57) than the lowest classes for $ECT_{w09}$ (0.53) or $ECT_{orig}$ (0.47), though this was only significant compared to $ECT_{orig}$ (p ≤ 0.05). In contrast, only marginal differences were noted when comparing stability classes 4 ($ECT_{new}$ 0.16, $ECT_{orig}$ 0.18).
Considering class 1 as an indicator of instability, the sensitivity was 0.44 with $ECT_{new}$ (0.58 when considering classes 1 and 2 together, Fig. 4c).

#### 4.3.2 Stability classification for two adjacent ECTs

70% of the time two ECTs indicated the same $ECT_{new}$ class, in 19% they differed by one class and in 11% by two (or more) classes.
Two ECTs resulting in the same $ECT_{new}$ class resulted in pronounced differences in the proportion of *unstable* slopes for classes 1 to 4 (0.61, 0.48, 0.20 and 0.13, respectively; Fig. 3c).
Randomly picking one of the two ECTs as the first ECT yielded the proportion *unstable* slopes as shown in Table 3. Additionally considering the outcome of a second ECT could increase or decrease the proportion *unstable* slopes for some combinations. For instance, if a first ECT resulted in either $ECT_{new}$ class 1 or 4, the second test would often indicate a similar result: class ≤ 2 in 85% of the cases, when the first ECT was class 1, and class ≥ 3 in 93% of the cases, when the first ECT was class 4. However, if the first ECT would either be $ECT_{new}$ class 2 or 3, a large range of proportion *unstable* slopes could result depending





**Table 3.** Proportion *unstable* slopes when randomly selecting one of two ECTs as the first test ($ECT_{new}(1^{st})$) (prop *unstable* $1^{st}$) and the number of cases (N) , and the respective proportion *unstable* slopes $2^{nd}$ following the outcome of the second ECT ($ECT_{new}(2^{nd})$).

| $ECT_{new}(1^{st})$ | prop *unstable* $1^{st}$ | N | $ECT_{new}(2^{nd})$ | N | prop *unstable* $2^{nd}$ |
|---|---|---|---|---|---|
| 1 | 0.54 | 129 | 1 or 2 | 110 | 0.6 |
|   |      |     | 3 or 4 | 19  | 0.18 |
| 2 | 0.48 | 49  | 1 or 2 | 35  | 0.54 |
|   |      |     | 3 or 4 | 14  | 0.32 |
| 3 | 0.19 | 98  | 1 or 2 | 21  | 0.27 |
|   |      |     | 3 or 4 | 77  | 0.17 |
| 4 | 0.14 | 175 | 1 or 2 | 13  | 0.19 |
|   |      |     | 3 or 4 | 162 | 0.13 |

on the second test result (0.54 - 0.17, Tab. 3), including some combinations resulting in the proportion *unstable* slopes being close to the base rate.

### 4.4 Comparison to Rutschblock test results

The proportion *unstable* slopes decreased significantly with each increase in RB stability class (0.76, 0.53, 0.25 and 0.11 for classes 1 to 4, respectively; $p < 0.01$; Fig. 3c). If a binary classification were desired, classes 1 and 2 would be considered as indicators of instability, classes 3 and 4 as relating to *stable* conditions. Employing this threshold, the sensitivity was 0.54 and the specificity 0.87 (Fig. 4d). Considering RB class 3, also termed «fair» stability (Schweizer, 2007), as an indicator of stability is, however, not truly supported by the data. This class has a proportion *unstable* slopes of 0.25, only marginally lower than the

base rate.

Comparing RB with the ECT showed that the proportion of *unstable* slopes for RB stability class 1 was significantly higher ($p < 0.01$) and for class 4 by about 0.05 lower ($p > 0.05$) than for any of the ECT classifications (Fig. 3a, c). This indicates that the RB stability classes at either end of the scale captured slope stability better than the ECT results, regardless which of the ECT classification was applied, and whether a second test was performed. Fig. 3a and c also highlight that RB class 2 and

ECT class 1 ($ECT_{w09}$, $ECT_{new}$) had similar proportions of *unstable* slopes. $ECT_{new}$ stability class 2 had a lower proportion of *unstable* slopes than RB class 2 ($p < 0.05$), but a higher proportion than RB class 3 ($p < 0.05$). The proportions of *unstable* slopes for the two highest $ECT_{new}$ classes were not significantly different than for the two highest RB classes ($p > 0.05$).

The false alarm rate of the RB (classes 1 and 2) was lower than for any of the ECT classifications (Fig. 4). However, in our data set a comparably large proportion of RB tests (0.34) indicated stability class 3 in slopes rated as *unstable*. This ratio is higher

than for single $ECT_{new}$ class 3. However, the frequency that stability class 4 (false *stable*) was observed in *unstable* slopes was lower than for $ECT_{new}$ class 4 (0.13 vs. 0.23, respectively).




The $ECT_{new}$ stability class correlated significantly with the RB stability class (Spearman rank-order correlation $\rho = 0.43$, p $<$ 0.001), a correlation which was stronger for ECT pairs resulting twice in the same ECT stability class ($\rho = 0.64$, p $< 0.001$).

### 4.5 The predictive value of stability tests

Now, we explore the predictive value of a stability test result as a function of the base rate, the proportion of *unstable* slopes. In our data set the proportion *unstable* slopes, the base rate, increased strongly with forecast danger level (1-Low: 0.02, 2-Moderate: 0.1, 3-Considerable: 0.38, Tab. 4).

Considering single $ECT_{new}$ class 1 and RB class 1 showed that PPV was always higher than the base rate (Fig. 5), indicating that the stability test predicted a higher probability for the slope to be *unstable* than just assuming the base rate. This shift was more pronounced for the Rutschblock than for the ECT, particularly at 1-Low and 2-Moderate. While PPV for stability class 1 (single or two ECT) remained low at 1-Low and 2-Moderate (PPV $\leq 0.3$, Tab. 4), indicating that it was still more likely that the slope was *stable* rather than *unstable*, the likelihood ratio indicated weak evidence in favor of instability (Tab. 4). At 4-High, the number of tests performed was very low (N = 16), therefore results are indicative at best.

Figure 5 also shows the shift in PPV, when considering $ECT_{new}$ or RB stability class 4 (high stability). In these slopes, PPV was lower than the base rate, indicating that the probability the specific slope tested to be *unstable* was less than the base rate. However, the resulting posterior probability was still higher compared to the base rate of the neighboring next lower danger level.

Analysing the entire data set together, regardless of the forecast danger level, the proportion *unstable* slopes was 0.21, and thus somewhat between the values for 2-Moderate and 3-Considerable. Again, the informative value of the test can be noted (Fig. 5). However, ignoring the specific base rate related to a certain danger level, leads - for instance - to an underestimation of the likelihood that the slope is *unstable* at 3-Considerable (RB or $ECT_{new}$ class 1), or an overestimation for the presence of instability at 1-Low (RB or $ECT_{new}$ class 4).

As shown in Figures 3c, the two extreme RB stability classes correlated better with slope stability than the respective two extreme $ECT_{new}$ classes. This is also reflected in Fig. 5 by the stronger shift from base rate to PPV, but can also be noted when calculating LR+ using a binary classification (LR+ for RB classes $\leq 2$ (25, 4.2, 3 for 1-Low, 2-Moderate, 3-Considerable) compared to single $ECT_{new}$ classes $\leq 2$ (5.2, 2.6, 2.9 for 1-Low, 2-Moderate, 3-Considerable)).

## 5 Discussion

### 5.1 Performance of ECT classifications

We compared ECT results, applying existing and testing a new classification with concurrent slope stability information. Quite clearly, whether a crack propagates across the entire column or not, is the key discriminator between *unstable* and *stable* slopes (Fig. 3b). This is in line with previous studies (e.g. Simenhois and Birkeland, 2006; Moner et al., 2008; Simenhois and

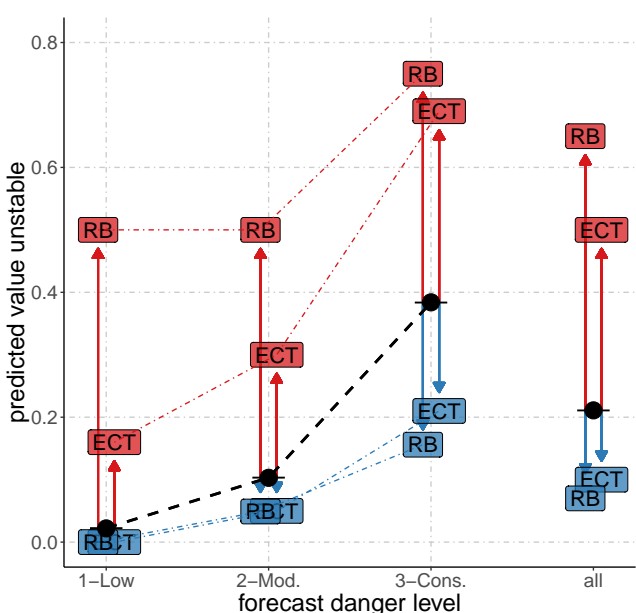

**Figure 5.** Positive predictive values (position of labels, RB - Rutschblock, ECT = single ECT$_{new}$) are shown compared to the respective base rate (black dots and black dashed line), the proportion of *unstable* slopes for danger levels 1-Low, 2-Moderate (2-Mod) and 3-Considerable (3-Cons), and for the entire data set (all). Predictive values are shown for the respective lowest (red colour, labels above base rate line) and highest (blue, labels below base rate line) stability classes. The arrows indicate the shift from the prior probability - the base rate at a given danger level - to the positive predictive value (posterior probability) for the specific slope tested being *unstable*.

**Table 4.** Positive predictive values (PPV) and positive likelihood ratios (LR+) for ECT$_{new}$ class 1 and classes 1 and 2 combined, stratified by forecast danger level (D$_{RF}$) and corresponding base rate proportion *unstable* slopes.

| | | | class 1 | | classes 1 + 2 | |
|---|---|---|---|---|---|---|
| D$_{RF}$ | N | prop *unstable* | PPV | LR+ | PPV | LR+ |
| 1-Low | 134 | 0.02 | 0.16 | 8.4 | 0.11 | 5.2 |
| 2-Moderate | 523 | 0.1 | 0.29 | 3.7 | 0.23 | 2.6 |
| 3-Considerable | 451 | 0.38 | 0.69 | 3.5 | 0.64 | 2.9 |
| 4-High | 16 | 0.44 | 1 | $+\infty$ | 0.53 | 1.4 |



Birkeland, 2009; Winkler and Schweizer, 2009; Techel et al., 2016) and with our current understanding of avalanche formation (Schweizer et al., 2008b). Moreover, our results confirm the proposition by Winkler and Schweizer (2009) that the number of
taps provides additional information allowing a better distinction between results related to *stable* and *unstable* conditions. The optimal threshold to achieve a balanced performance, i.e. high sensitivity as well as high specificity, was found to be between ECTP20 and ECTP22, depending on the method (*kmeans*-clustering, *pROC*-cutoff point). This finding agrees well with the threshold proposed by Winkler and Schweizer (2009) who suggested ECTP21. Using the binary classification, as originally proposed by Simenhois and Birkeland (2009), increased the sensitivity but led to a rather high false alarm rate. Moving away
from a binary classification increased PPV and NPV for the lowest and highest stability classes, respectively, but came at the cost (or benefit) of introducing intermediate stability classes.

Only in some situations did pairs of ECTs performed in the same snow pit show an improved correlation with slope stability: when two tests were either $ECT_{new}$ stability class 1 or 2, or when either both tests were class 4, or one class 3 and one class 4.

### 5.2 Comparing ECT and Rutschblock

To our knowledge, and based on the review by Schweizer and Jamieson (2010), there have only been three previous studies which compared ECT and RB in the same data set.

Moner et al. (2008), in the Spanish Pyrenees, relying on a comparably small data set of 63 RB (base rate 0.44) and 47 single ECT (base rate 0.38) observed a higher unweighted average accuracy for the ECT (0.93) than the RB (0.88). In contrast, Winkler and Schweizer (2009, N = 146, base rate 0.25) presented very similar values for RB (0.84) and the ECT (0.81).
However, Winkler and Schweizer (2009) partially relied on a slope stability classification which is based strongly on the Rutschblock. Therefore, they emphasized that the RB was favored in their analysis. And finally, the data presented by Techel et al. (2016) is to a large part incorporated in the study presented here.

In that respect, this study presents the first comparison incorporating a comparably large number of ECT and RB conducted in the same snow pit, where slope stability was defined independently of test results. Seen from the perspective of the proportion
*unstable* slopes, the lowest and highest RB classes correlated better with slope stability than the respective ECT classes. Incorporating the sensitivity, the proportion of *unstable* slopes detected by a test, a mixed picture showed: Single ECT and RB (classes 1 and 2) detected a comparable proportion of *unstable* slopes (0.58 vs. 0.53, respectively, Fig. 4c, d). False-unstable classifications, however, were comparably rare for the RB (0.12) compared to single ECT (0.23). In other words, a RB detected less reliably an *unstable* slope than an ECT, because intermediate RB results were still rather frequent in these slopes. At
the same time, RB results indicating stability on *unstable* slopes were less frequent than ECT indicating stability (RB: 0.13, ECT: 0.23). However, when a RB test indicated instability, this provided stronger evidence that the slope was in fact *unstable* compared to an ECT indicating instability, as the latter were much more frequently also observed on *stable* slopes.

### 5.3 On the predictive value of stability tests

We recall the three lessons drawn by Ebert (2018) in his theoretical investigation of the predictive value of stability tests using
Bayesian reasoning in avalanche terrain, as this greatly inspired us to explore these aspects using actual observations and



compare them to our results:

(1) «A localised diagnostic test will be more informative the higher the general avalanche warning.» (Ebert, 2018, p. 4). With general «avalanche warning» , Ebert (2018) refers to the forecast danger level as a proxy to estimate the base rate. As shown in Fig. 5, PPV increased for both ECT and RB with increasing base rate / danger level, supporting this statement. From a more theoretical perspective, it can be shown that PPV can be derived from Bayes Theorem (e.g. Blume, 2002; Ebert, 2018), therefore linking both approaches.

(2) «... Do not 'blame' the stability tests for false positive results: they are to be expected when the avalanche danger is low. In fact, their existence is a consequence of the basic fact that low-probability events are difficult to detect reliably» (Ebert, 2018, p. 4). Fig. 5 supports this statement: at 1-Low and 2-Moderate an ECT indicating instability was much more often observed on a *stable* slope rather than an *unstable* one. Only once the base rate was sufficiently high, in our case at 3-Considerable, tests indicating instability were observed more often on *unstable* rather than *stable* slopes.

(3) «In avalanche decision-making, there is no certainty, all we can do is to apply tests to reduce the risk of a bad outcome, yet there will always be a residual risk» (Ebert, 2018, p. 5). The likelihood ratio was greater than 1 for tests indicating instability, regardless whether we considered an ECT or a RB result and regardless of the danger level, and less than 1 for tests indicating stability. This is statistical evidence for a higher probability that a slope is *unstable* compared to the base rate. From a Bayesian perspective, we would say that a positive test (a low stability class) always increases our belief that the slope is *unstable*, and vice versa when a test is negative (a high stability class).

In summary, and regardless of the strength of evidence, instability tests are useful despite the uncertainty which remains.

### 5.4 Sources of error and uncertainties

Beside potential misclassifications in slope stability, which we address more specifically in the following section (Sect. 5.5), Schweizer and Jamieson (2010) pointed out two other sources of error. The first of these is linked to the test method, which are relatively crude methods and where, for instance, the loading may vary depending on the observer. The second error source is linked to the spatial variability of the snowpack. The constellation of slab and underlying weak layer varies in the terrain and may consequently have an impact on the test result. Furthermore, this data set did not permit to check whether the failure plane of avalanches or whumpfs was linked to the failure plane observed in test results. Such information about the «critical weak layer» was, for instance, incorporated by Simenhois and Birkeland (2009) and Birkeland and Chabot (2006) in their analyses. However, from a stability perspective, considering the actual test result is the more relevant information.

### 5.5 The influence of the reference class definitions and the base rate

So far we have explored ECT and RB assuming that there are no misclassifications of slope stability. However, as the true slope stability is often not known (particularly in stable cases), errors in slope stability classification will occur. Such errors, however, may potentially influence all the statistics derived to describe the performance of tests (Brenner and Gefeller, 1997). For instance, if there are at least some slopes misclassified, classification performance will drop. However, in such cases, POD and PON will additionally be influenced by the true (though unknown) base rate (Brenner and Gefeller, 1997).





In previous studies exploring ECT (Moner et al., 2008; Simenhois and Birkeland, 2009; Winkler and Schweizer, 2009), slope

stability classifications were generally well described and the base rate for the applied slope stability classification given. However, slope stability classification approaches differed somewhat. For instance, a stability criterion used by Moner et al. (2008) was the occurrence of an avalanche on the test slope, while Simenhois and Birkeland (2009) additionally considered explosives-testing of the slope as relevant information. Winkler and Schweizer (2009), on the other hand, additionally considered the manual profile classification used operationally in the Swiss avalanche warning service (Schweizer and Wiesinger,

2001; Schweizer, 2007) and considered a sufficient criterion for instability, when profiles were rated as «very poor» or «poor». As this classification relies rather strongly on the RB result, the RB would be favored in such an analysis (Winkler and Schweizer, 2009).

We have no knowledge about the uncertainty linked to our classification. However, we can demonstrate the impact of variations in the definition of the reference class on summary statistics like POD and PON, and using different data subsets for

analysis: Let us assume we are not interested in comparing ECT and RB, but want to explore only the performance of a binary ECT classification with ECTP22 as the threshold between two classes. We will, however, use the RB together with the criteria introduced in Section 2.3 to define slope stability:

- Without using the RB as an additional criteria, POD and PON for the ECT was 0.58 and 0.77, respectively (Fig. 4c).

- If only slopes are considered *unstable*, when the RB stability class was $\leq 2$, and those as *stable* with RB stability class

$\geq 3$, the resulting POD is 0.70 and PON is 0.84. The base rate in this data set is 0.14 and N = 591.

- Being even more restrictive, and considering only slopes *unstable*, when the RB stability class was 1, and those as *stable* with RB stability class 4, the resulting POD is 0.75 and PON is 0.89. The base rate in this data set is 0.14 and N = 294.

Of course, one could also be interested in exploring the performance of the RB, and define slope stability by using ECT results as additional criteria to those in Section 2.3. Without relying on ECT results, POD and PON for the RB were 0.54 and 0.87,

respectively (Fig. 4d). Considering $ECT_{new}$ stability class $\leq 2$ as *unstable*, else as *stable*, POD and PON would increase to 0.66 and 0.91 (N = 561), or 0.71 and 0.93, respectively when considering only $ECT_{new}$ stability class 1 as *unstable* and class 4 as *stable* (N = 385).

The combination of various error sources (Sect. 5.4), together with varying definitions of slope stability and differences in the base rate make it almost impossible to directly compare results obtained in different studies. Therefore, performance values

presented in this study, but also in other studies regarding snow instability tests, must always be seen in light of the specific data set used and allow primarily a comparison within the study.

## 6 Conclusions

We explored a large data set of concurrent RB and ECT, and related these to slope stability information. Our findings confirmed the well-known fact that crack propagation propensity, as observed with the ECT, is a key indicator relating to snow instability.

In addition, the number of taps required to initiate a crack also provides information concerning snow instability. Combining

crack propagation propensity and the number of taps required to initiate a failure allows refining the original binary classification. We propose an ECT stability interpretation with four distinctly different stability classes. Furthermore, for an ECT result being in one of the two intermediate classes, a second ECT performed in the same snow pit may be the decisive factor towards either the highest or lowest stability class that are best related with rather *stable* or *unstable* conditions, respectively. In our

data set, the proportion of *unstable* slopes was higher and lower in the lowest and highest stability class for the RB than for the ECT. Hence, the RB correlated better with slope stability than the ECT.

We discussed further that changing the definition of the reference standard, the slope stability classification, has a large impact on summary statistics like POD or PON. This hinders comparison between studies, as differences in study designs, data selection and classification must be considered.

And finally, we investigated the predictive value of stability test results using a data-driven perspective. We conclude by rephrasing Blume (2002): When a stability test indicates instability, this is always statistical evidence for instability, as this will increase the likelihood for instability compared to the base rate. However, in case of a low base rate, false unstable predictions are likely.

*Author contributions.* FT designed the study, extracted and analyzed the data, and wrote the manuscript. MW extracted and classified a

large part of the text from the snow profiles. KW, JS and AvH provided in-depth feedback on study design, interpretation of the results and manuscript.

*Competing interests.* No competing interests.

*Acknowledgements.* REVIEWERS



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
