# Peer review of "On snow stability interpretation of Extended Column Test results"

_Natural Hazards and Earth System Sciences, 2020_

## Referee Comment (RC1) · Bret Shandro (Referee) · 19 Apr 2020

**General comments**

This manuscript presents a novel method for interpreting snowpack tests for evaluating snow avalanche hazard and is appropriate for the NHESS. Overall, the quality of the manuscript is good to excellent. The presentation of a 4-class stability interpretation scheme is beneficial beyond academic purposes, as some avalanche practitioners assess an avalanche problem's sensitivity on a 4-class scale (Statham et al., 2018). Below I proved minor comments for the authors and editor and recommend publication of the manuscript.

[Figure]

As the NHESS audience includes readers beyond snow avalanche hazard, I suggest a title that communicates the relevant natural hazard, for example, "On the snowpack stability interpretation of extended column test results."

**Specific comments**

Line 105 – Regarding the minimal depth criteria, Techel and Pielmeier (2014) appear to use a 15 cm. What is the benefit of distinguishing between a weak layer 6-10 cm and 5cm or less? Why not classify all tests class 4 if the weak layer less than 10cm?

Line 146 – For the dataset sampling to cluster stability classes, were any precautions taken to avoid the algorithm producing results that were overfitted to the sampled data, i.e. how was a 90-10 ratio selected?

Figure 3 – The reader may benefit from the proportion values included in the figure. I believe this would allow the reader to better interpret the results section.

**Technical corrections**

Line 168 – There appears to be a formatting issue with the list, (i) (ii).

**References**

Statham, G., Haegeli, P., Greene, E., Birkeland, K., Israelson, C., Tremper, B., Stethem, C., McMahon, B., White, B., Kelly, J. (2018). A conceptual model of avalanche hazard. Natural Hazards, 90(2), 663–691. https://doi.org/10.1007/s11069-017-3070-5

Techel, F. and Pielmeier, C.: Automatic classification of manual snow profiles by snow structure, Nat. Hazards Earth Syst. Sci., 14, 779–787, https://doi.org/10.5194/nhess-14-779-2014, 2014.
* * *

---

## Referee Comment (RC2) · Markus Landrø (Referee) · 20 Apr 2020

General comments. In this manuscript the authors present a new stability interpenetration scheme for the Extended Column Test. The test is commonly used among observers associated with forecasting services, as well as back-county recreationalist. This work addresses a relevant and important topic within operational avalanche forecasting. The findings can be of interest for avalanche forecasting services worldwide. I found it well written and structured, generally very clear and correct language. Figures are of good quality. I recommend publishing the manuscript. I only have minor suggestions on how to improve the paper.

Specific comments p1 l21 what about the risk involved?         p1 l22

what about the radar on skies initiative. Could you comment on that https://sknow.ski/?fbclid=IwAR180DSVe2nRwPwOfDM6b73niDjzB4uLgbTk6i3c3Smn2H_ZC_sfq1j3Aaw p4 l99: consider adding fatal skier-triggered avalanches. p14 l345-358 is there a difference in test performance dependent on weak layer properties (grain type, grain size, weak layer thickness). you probably have this data from the test sites. It could also be interesting if you related it to forecasted avalanche problem. p4 l102-105: what if the overlaying snow is harder than lets say 1F. Does that have an impact? not theme of this paper, but still.. p17 5.4-5.5 consider also relating it to avalanche problems which have become an important part of avalanche forecasting. What when you have low probability and high consequence. i.e. deep persistent weak layer. Another challenge is that on a day of back-country touring you will probably seek the most stable conditions whereas observers will seek the most unstable areas to perform their tests. Especially in situations where you don't have any signs of instability this can possibly bias you slope stability classification in addition to the other sources of error you included. You have not addressed vertical vs lateral tapping and the energy absorption due to deformation of the upper snow layers above the weak layer. https://arc.lib.montana.edu/snow-science/item/2673

Technical corrections: p1 l2: consider changing into to in p2 l33: consider changing to improve with improving p2 l50: remove comma after Both p3 l72: insert The test procedure p4 l104: remove comma after (2014) p5 l116: remove comma after stable p5 l117: consider changing relates to relate p6 l145: change were to was p6 l154: consider changing its with it's p7 l177: add The probability P9 l218: consider removing comma after slopes p9 l226: consider changing was to were p9 l233: consider adding proportion of p10 l242: consider adding Regardless of p10 figure text: consecutive numbers p16 l376: consider adding one or two commas , in fact, p18 l420: consider removing comma instability, p18 l439: consider changing make to makes p18 l440: consider removing in p18 l445: consider changing in addition to Also p19 l449: consider changing are best to is best p19 l457: change for to of

---

## Short Comment (SC1) · 24 Apr 2020

Due to the high variability of snow, it is nearly impossible for instability tests to accurately predict stable or unstable conditions 100% of the time. However, instability tests have proven to be a useful avalanche forecasting tool for both backcountry recreationalists and avalanche forecasters.

Classifying instability test results is the trickiest part of stability assessment. This paper explores a new instability classification for the Extended Column Test (ECT). The ECT is the most widely used instability test by backcountry recreationalists and practitioners. Snowpilot shows that roughly 77% of snowpit profiles with instability tests have at least one ECT (Snowpilot data 2019).

[Figure]

While this paper does an overall good job exploring different test result classifications for the ECT, there is room for improvement. One flaw is comparing ECT classifications with Rutschblock (RB) classifications. The RB was the first quantifiable instability test that assessed both initiation and propagation on a specific slab/weak layer combination. The RB has proven to be an effective test, however, it is gradually losing popularity with backcountry recreationalists due to its cumbersome nature. Snowpilot shows that over the past ten seasons, the RB has been entered as a test result into less than 1% of snowpit profiles that have at least one instability test. This is significantly lower than the ECT which is entered into 77% of profiles with instability tests.

Because the ECT has nearly replaced the RB, more emphasis should be placed on how to interpret the ECT effectively without correlating it to the RB. Currently, the ECT works on a binary classification scale, stable or unstable. An ECTP test result under standard loading steps is considered an unstable result. A test result of ECTN or ECTX is considered a stable result.

It does make sense to integrate a number classification with a specific ECT test result, but this paper presents number classifications in a way that may be confusing to backcountry practitioners. This paper suggests that a Class 1 stability rating indicates low stability or mostly unstable conditions. This classification is confusing in two ways. One, it uses the word low stability to indicate unstable conditions, possibly Considerable to High danger. Backcountry practitioners could potentially associate low stability with a low danger, which would be a dangerous interpretation. Second, backcountry practitioners could potentially associate Class 1 (unstable conditions) with Level 1 on the North American Danger Scale which is a Low danger or generally stable conditions.

Changing the number classifications around so that Class 4 predicts highly unstable conditions and Class 1 predicts stable conditions would be more consistent with the number classifications of the current North American Avalanche Danger Scale. Combining Class 2 and Class 3 to form an intermediate classification could present interpretation problems. Replacing the word intermediate with the word moderate may

lead to easier interpretation. One possible solution could be a four-level classification scheme that follows the current danger scale. Example: Class 1) Mostly Stable Conditions ECTN or ECTX, Class 2) Moderately Unstable Conditions ECTP >_21, Class 3) Considerably Unstable Conditions ECTP 12-20, Class 4) Highly Unstable Conditions ECTP<_11.

Avalanche education and snow science are fluid practices and examining different ways to classify instability test results will ultimately help us better understand snow stability and avalanches. Developing a numerical classification scheme for the ECT seems like a logical step in expanding the current binary classification of stable or unstable. From a practitioner standpoint, aligning the classification numbers with the current North American Danger Scale could make for easier interpretation. In regard to the Rutschblock, data suggests that backcountry recreationalists and practitioners are rarely using the RB as a consistent instability test. Due to the low % of RB's being conducted in the field, focusing on the development of a more thorough ECT classification is worthwhile.

Please contact me with any questions or for clarification on any items comment on above. Thank you for your time and effort.

Eric Knoff Six Points Avalanche Education info@avalancheclass.com 307-690-3898
* * *
[Figure]

**Snopilot Data**
**15,540 snowpits with stability tests**

| Code: | CT | ECT | RB | SB | ST | PST | Total |
|---|---|---|---|---|---|---|---|
| 2010 | 411 | 426 | 9 | 2 | 52 | 77 | 555 |
| 2011 | 590 | 630 | 12 | 6 | 88 | 110 | 842 |
| 2012 | 477 | 459 | 17 | 1 | 85 | 89 | 658 |
| 2013 | 364 | 403 | 7 | 1 | 29 | 81 | 525 |
| 2014 | 391 | 457 | 5 | 2 | 53 | 97 | 605 |
| 2015 | 339 | 472 | 5 | 7 | 33 | 82 | 600 |
| 2016 | 1707 | 1926 | 25 | 14 | 177 | 324 | 2555 |
| 2017 | 2879 | 3436 | 17 | 9 | 301 | 574 | 4363 |
| 2018 | 3382 | 3691 | 30 | 6 | 378 | 583 | 4777 |
| 2010 | 74.05% | 76.76% | 1.62% | 0.36% | 9.37% | 13.87% | 100.00% |
| 2011 | 70.07% | 74.82% | 1.43% | 0.71% | 10.45% | 13.06% | 100.00% |
| 2012 | 72.49% | 69.76% | 2.58% | 0.15% | 12.92% | 13.53% | 100.00% |
| 2013 | 69.33% | 76.76% | 1.33% | 0.19% | 5.52% | 15.43% | 100.00% |
| 2014 | 64.63% | 75.54% | 0.83% | 0.33% | 8.76% | 16.03% | 100.00% |
| 2015 | 56.50% | 78.67% | 0.83% | 1.17% | 5.50% | 13.67% | 100.00% |
| 2016 | 66.81% | 75.38% | 0.98% | 0.55% | 6.93% | 12.68% | 100.00% |
| 2017 | 65.99% | 78.75% | 0.39% | 0.21% | 6.90% | 13.16% | 100.00% |
| 2018 | 70.80% | 77.27% | 0.63% | 0.13% | 7.91% | 12.20% | 100.00% |

**Fig. 1.**

---

## Author Comment (AC1) · 1 May 2020

Dear Eric

Thank you for your interest in this study and for your comments regarding the content of the manuscript, which we greatly appreciate. In the following, we reply on the main points you raised.

We are aware that the ECT has become (one of) the most widely used instability tests used by practitioners in the world. Even in Switzerland, where the Rutschblock is still the standard stability test performed by field observers, the ECT has gained a lot in popularity. This shows when looking at the numbers of profiles entered in

our data-base with at least one ECT: profiles with an ECT are now almost equal in numbers compared to profiles with a RB (often both tests are performed). Exactly this increasing popularity of the ECT was our motivation to revisit the existing stability classifications.

However, we strongly disagree with your statement that just because the RB is not as popular anymore, it is a "flaw" of the study to compare RB and ECT. On the contrary, we consider this a strength of our study, namely that we are able to perform this comparison. Only this comparison of the tests performed in the same snow pit allows to compare the performance of both tests.

We have deliberately not assigned class labels yet. Instead, for the purposes of this manuscript, we introduced the class numbers to be able to assign a clear increasing order to the classes (Section 2.2 on page 4). For a better distinction between test stability classes and slope stability ratings in the text of the manuscript, we introduced low and high stability, as we decided to use the terms unstable and stable for slope stability. We called the middle class intermediate, as - at this stage in the manuscript - we only knew that it was between the classes indicating more often unstable or stable conditions according to Winkler and Schweizer (2009). Furthermore, we disagree with your recommendation that the four class numbers and labels should line up with the danger levels. This might suggest that a single test result correlates with the danger level, which is obviously not the case. It is not possible to deduce a danger level from a single stability test (e.g. Schweizer et al., 2003). In fact, we will often observe a whole range of stability test results on a day with, for instance, Moderate or Considerable danger.

We think any labels should follow the established labeling for snow stability, which includes the main classes: poor, fair, good. Hence, to rate the ECT results we suggest the following three (primary) stability class labels, with poor being additionally split into two sub-classes:

- ECTN and ECTX: **good** (which includes cases of very good)

- ECTN<10 or ECTP>21: **fair**

- ECTP14 to ECTP21: **poor** (tending to fair)

- ECTP<14: **poor** (including cases of very poor)

These class labels are close to the stability rating scheme used in the operational guidelines in the U.S., Canada and Switzerland. Furthermore, they permit alignment of ECT and RB somewhat, with very poor stability remaining for RB results with the most unstable stability class. In the end, the snow safety community will decide, whether a three- or four-class classification scheme is suitable, and what their labels will be. Our aim is to provide the basis for an informed decision, grounded in data.

On behalf of the authors,

Frank Techel (techel@slf.ch)

---

## Author Comment (AC2) · 1 May 2020

**Reply to reviewer comments by Markus Landrø**

Frank Techel

*Correspondence to:* Frank Techel (techel@slf.ch)

Dear Markus

Thank you very much for reviewing our manuscript and the helpful points regarding improvements.

Please find below a point-by-point reply to your specific comments (your comments in *blue*, our reply **black**):

*Specific comments*

*p1 l21 what about the risk involved?*

The safety of the person performing a test is, obviously, very important when selecting a suitable site. - We will add a statement in this regard.

*p1 l22 what about the radar on skies initiative. Could you comment on that https:// sknow.ski/ ?fbclid=IwAR180DSVe2nRwPwOfDM6b* *ZC_sfq1j3Aaw*

No, we are unable to comment on this initiative, as we don't know anything about it other than what it says on the website you mention.

*p4 l99: consider adding fatal skier-triggered avalanches.*

The two studies (Schweizer and Lütschg, 2001; van Herwijnen and Jamieson, 2007), which we have cited, explored skier-triggered avalanches in general, not just fatal skier-triggered avalanches.

*p14 l345-358 is there a difference in test performance dependent on weak layer properties (grain type, grain size, weak layer thickness). you probably have this data from the test sites. p4 l102-105: what if the overlaying snow is harder than lets say 1F. Does that have an impact? not theme of this paper, but still. It could also be interesting if you related it to the forecasted avalanche problem.*

In this study, we did not explore the role of the snowpack structure and layering on the test results. However, Winkler and Schweizer (2009), who compared snow stability tests like ECT and RB, also analyzed in detail the respective properties of the failure layers and the slab overlying the failure interface on stable and unstable slopes. They noted that failure layer hardness, failure layer grain type, failure layer grain size and differences in hardness across the failure interface were

significant variables distinguishing between the ECT and the RB failure planes on stable and unstable slopes. For more details, please refer to their article.

*p17 5.4-5.5 consider also relating it to avalanche problems which have become an important part of avalanche forecasting. What when you have low probability and high consequence. i.e. deep persistent weak layer. Another challenge is that on a day of back-country touring you will probably seek the most stable conditions whereas observers will seek the most unstable areas to perform their tests. Especially in situations where you don't have any signs of instability this can possibly bias your slope stability classification in addition to the other sources of error you included.*

Indeed, there is a different focus when undertaking a back-country tour, with the goal to find the best skiing in stable conditions, and when finding a suitable location to perform a stability test (which are often performed in locations where snowpack is thinner, and therefore likely weaker).

More specifically regarding our study: We don't know whether a snow pit location represented the surrounding terrain well. It is one of the potential error sources, which may influence the quality of our slope stability classification. While this is currently addressed in a rather general way in the statement on lines 402-404, we will add a more specific comment in this regard.

*You have not addressed vertical vs lateral tapping and the energy absorption due to deformation of the upper snow layers above the weak layer. https://arc.lib.montana.edu/snow-science/item/2673*

We have not compared vertical vs lateral tapping, as both tests are loaded vertically from the top. Therefore, we do not discuss these.

*Technical comments*

Thank you for pointing these out, we will address them as suggested.

**References**

Schweizer, J. and Lütschg, M.: Characteristics of human-triggered avalanches, Cold Reg. Sci. Technol., 33, 147–162, doi:10.1016/s0165-232x(01)00037-4, 2001.

van Herwijnen, A. and Jamieson, B.: Snowpack properties associated with fracture initiation and propagation resulting in skier-triggered dry snow slab avalanches, Cold Regions Science and Technology, 50, 13–22, doi:https://doi.org/10.1016/j.coldregions.2007.02.004, 2007.

Winkler, K. and Schweizer, J.: Comparison of snow stability tests: Extended Column Test, Rutschblock test and Compression Test, Cold Regions Science and Technology, 59, 217–226, doi:10.1016/j.coldregions.2009.05.003, 2009.

---

## Referee Comment (RC3) · Markus Landrø (Referee) · 4 May 2020

P4 L99. I re-read the sentence, and it is ok. (rarely associated). In my experience, in areas with very strong winds, dry and cold climate, this is not so rare, but I have no study to underpin this.

Regarding the vertical vs lateral tapping. My point is to encourage you to discuss the limitations of the ECT even more. Energy absorption is one thing. What about the inconsistency in the tapping itself. See the attached illustration.
* * *
**Fig. 1.**

---

## Author Comment (AC3) · 5 May 2020

Thank you for posting this interesting graph, which highlights the variations between operators performing the ECT. Unfortunately, these findings do not seem to be published. However, there seems to have been a similar analysis more than ten years ago for the Compression Test (CT) by Spencer Logan (2008: *Are you a hard hitter? Systematic measurement error in the compression test*) https://arc.lib.montana.edu/snow-science/objects/issw-2006-682.pdf Unfortunately, only an abstract without any results is available.

We will take up your suggestion to discuss the limitations of the ECT in greater depth.

2020-50, 2020.

---

## Author Comment (AC4) · 23 May 2020

**Reply to reviewer comments by Bret Shandro**

Frank Techel

*Correspondence to:* Frank Techel (techel@slf.ch)

Dear Bret

thank you very much for your review of our manuscript and the helpful comments.

Please find below our reply (in blue) to your comments *(in italics)*.

*General comments*

*This manuscript presents a novel method for interpreting snowpack tests for evaluating snow avalanche hazard and is appropriate for the NHESS. Overall, the quality of the manuscript is good to excellent.-* Thank you for this very positive feedback.

*The presentation of a 4-class stability interpretation scheme is beneficial beyond academic purposes, as some avalanche practitioners assess an avalanche problem's sensitivity on a 4-class scale (Statham et al., 2018).*

*Below I proved minor comments for the authors and editor and recommend publication of the manuscript.*

*As the NHESS audience includes readers beyond snow avalanche hazard, I suggest a title that communicates the relevant natural hazard, for example, «On the snowpack stability interpretation of extended column test results.» -* We intend to change the title as suggested.

*Specific comments*

– *Line 105 – Regarding the minimal depth criteria, Techel and Pielmeier (2014) appear to use a 15 cm. What is the benefit of distinguishing between a weak layer 6-10 cm and 5 cm or less? Why not classify all tests class 4 if the weak layer less than 10 cm? -* The idea was a less discrete influence of the weak layer depth on the classification. However, comparing the results using a simpler approach as you suggest with the approach we used in the manuscript, showed only very marginal changes in the results. As keeping it simple has some benefit too, we will adjust the weak layer criteria to a single criteria: a depth less than 10 cm will be classified as stability class 4. Using this simpler depth criteria will have no impact on the overall findings or conclusions drawn (despite some minor changes in proportion values in parts of the manuscript, which we will address in the revised mansuscript).

- *Line 146 – For the dataset sampling to cluster stability classes, were any precautions taken to avoid the algorithm producing results that were overfitted to the sampled data, i.e. how was a 90-10 ratio selected?* - Although not shown in the manuscript, we also explored a sampling approach using an 80-20 ratio. The resulting splits were very similar as can be seen in Fig. 1. The most notable difference in the splitting criteria were noted for the class threshold between classes 3 and 4. Here, the first splits differed ($ECTN \leq 10$ vs. $ECTN \leq 3$). However, the second most frequent split obtained with 80% of the data ($ECTN \leq 10$) was the same as the most frequent split obtained with 90% of the data. - Note there is a mistake in the manuscript on line 260 which should read: *$ECTP \leq 14$ (48%), $ECTP \leq 13$ (36%)* rather than *$ECTP \leq 15$ (48%), $ECTP \leq 14$ (36%)*.

- *Figure 3 – The reader may benefit from the proportion values included in the figure. I believe this would allow the reader to better interpret the results section.* - Good suggestion. We will add these.

*Technical corrections*

- *Line 168 – There appears to be a formatting issue with the list, (i) (ii).*

[Figure]

**Figure 1.** Clustering thresholds obtained, when using either 90% (currently used in the manuscript) or 80% of the data for each of the 100 repetitions. Colours represent the four classes based on the most frequently indicated splitting criteria. The dotted-dashed lines indicate the second most frequent splitting criteria. In general, the splitting criteria were rather similar.

**References**

Statham, G., Haegeli, P., Greene, E., Birkeland, K., Israelson, C., Tremper, B., Stethem, C., McMahon, B., White, B., and Kelly, J.: A conceptual model of avalanche hazard, Natural Hazards, 90, 663 – 691, doi:10.1007/s11069-017-3070-5, 2018.

Techel, F. and Pielmeier, C.: Automatic classification of manual snow profiles by snow structure, Nat. Hazards Earth Syst. Sci., 14, 779–787, doi:10.5194/nhess-14-779-2014, 2014.

---

## Author Response (AR1)

**List of most important changes**

Reviewer 1 proposed a simpler depth criterion. We have adjusted the manuscript in the Methods section (l. 101-102). As a consequence some of the results presented changed in a rather minor way (none of the key findings nor the interpretation were affected by this change!)

Feedback received by Philip Ebert in a private email (we cited P. Eberts' study) suggested to rephrase the description of PPV and NPV and the proportion of unstable slopes as the original description was not fully clear. We have adjusted the manuscript accordingly (l. 185-198).

Feedback received by Philip Ebert suggested to show data for both stability tests in Table 4. We added this data. To address the data shown in this expanded table, we added (or rephrased) several lines (l. 337-340, 342-345).

Feedback by Eric Knoff (public discussion) suggested to introduce class labels rather than class numbers. We added a short section in this regard (Section *5.6 Proposing stability class labels* , l. 451-464) and a new Figure 6 visualizing classes together with class labels for the two tests.

**Point-by-point response to reviews**

Please find below a point-by-point response to the reviews indicating the respective line numbers in the original and the revised manuscript. Please also refer to the manuscript showing all the track changes for further changes, and to the replies to the reviewer comments on the discussion site.

**Reviewer #1 – Bret Shandro:**

| Original version | Revised manuscript |
| --- | --- |
| As the NHESS audience includes readers beyond snow avalanche hazard, I suggesta title that communicates the relevant natural hazard, for example, "On the snowpackstability interpretation of extended column test results." | We changed the title to: *On the snow stability interpretation of Extended Column Test results* |
| 105 – Regarding the minimal depth criteria, Techel and Pielmeier (2014) appear to use a 15 cm. What is the benefit of distinguishing between a weak layer 6-10 cm and 5 cm or less? Why not classify all tests class 4, if the weak layer less than 10 cm? | We have addressed this issue by simplifying the criterion to (101-102): *We addressed this by assigning stability class 4 if the failure layer was less than 10 cm below the snow surface.* See also reply to reviewer. Please note, this simplification of the criterion had a minor effect on some of the results shown. However, none of the key findings (or their interpretations) were affected by this change. Please refer to the track-changes-version of the revised manuscript, where these changes are highlighted. |
| 146 – For the dataset sampling to cluster stability classes, were any precautions taken to avoid the algorithm producing results that were overfitted to the sampled data, i.e. how was a 90-10 ratio selected? | We added a line in that respect (253-254): *Applying the same approach with 80% of the data (rather than with 90%) resulted in very similar class thresholds (LINK TO SUPPLEMENT).* We will provide a link to a supplement, which will be an extract of our reply to the reviewer. |
| Figure 3 – The reader may benefit from the proportion values included in the figure. I believe this would allow the reader to better interpret the results section. | We have added the proportion values in Figure 3a and 3c. |
| 168 – There appears to be a formatting issue with the list, (i) (ii). | We changed the formatting to (a) and (b). |

**Reviewer #2 – Markus Landrø:**

| Original version | Revised manuscript |
| --- | --- |
| 21: what about the risk involved? | 19: *changed to* Furthermore, considerable experience in the selection of a representative *and safe* site is needed, and the interpretation of test results is challenging. |
| 99: consider adding fatal skier-triggered avalanches | Not addressed, see reply to reviewer. |
| 345-358: is there a difference in test performance dependent on weak layer properties (grain type, grain size, weak layer | Not addressed, see reply to reviewer. |

| | |
|---|---|
| thickness). You probably have this data from the test sites. | |
| 102-105: what if the overlaying snow is harder than lets say 1F. Does that have an impact? Not theme of this paper, but still. It could also be interesting if you related it to the forecasted avalanche problem. | Not addressed, see reply to reviewer. |
| 2: consider changing into to in | |
| 33: consider changing to improve with improving | 31: done |
| 50: remove comma after Both | 48: done |
| 72: insert The test procedure | |
| 104: remove comma after (2014) | |
| 116: remove comma after stable | |
| 117: consider changing relates to relate | 213: changed from singular to plural |
| 145: change were to was | 141: done |
| 154: consider changing its with it's | |
| 177: add The probability | |
| 218: consider removing comma after slopes | |
| 226: consider changing was to were | 213: changed from plural to singular |
| 233: consider adding proportion of | 220: done |
| 242: consider adding Regardless of | |
| p10 figure text: consecutive numbers | Fig. 3: done |
| 376: consider adding one or two commas , in fact, | |
| 420: consider removing comma instability, | |
| 439: consider changing make to makes | |
| 440: consider removing in | |
| 445: consider changing in addition to Also | |
| 449: consider changing are best to is best | |
| 457: change for to of | 464: done |

We did not address the other suggestions, as we believe that the grammar was correct as it was.

**Public – Eric Knoff:**

| Original version | Revised manuscript |
|---|---|
| Eric Knoff proposed to use (or introduce) class labels rather than class numbers. – see also the detailed reviewer comment | We have taken up this suggestion. In that respect, we added a new subsection (Section 5.6) together with the new Figure 6 (part of which was already shown in . See also our detailed reply to Eric Knoff. |

**Additional Feedback – P. Ebert received via email:**

| Original version | Revised manuscript |
|---|---|
| 189-194: The formula and the description don't match up. The proportion of "unstable slopes" is most naturally understood as: a+c/ a+b+c+d. (the number of unstable slopes/ total number of slopes), which in effect is just the base rate. | We have taken up this comment, and rephrased accordingly with the goal to make it more easily understandable what we mean, when we refer to the *proportion of unstable slopes*. Please refer to lines 182-193. |

| | |
|---|---|
| a/a+b is, I think, best characterised as the proportion of "correct unstable predictions" (a= number of correct unstable slope prediction/ a+b= total number of predictions indicating unstable slope) or the proportion of unstable slopes, given the test results instability (as you say above). It's just that shortening it, and saying "proposition of unstable slope" doesn't make it a proportion on predictions but a proportion on facts, and so it is possibly misleading.
Also, the same applies to the NPV value:

d/c+d is the proportion of correct stable prediction, but not proportion of "stable slopes". The latter is
b+d/a+b+c+d | |
| Table 4: It be nice to have a table here for the RB as well. | We agree that this information would be beneficial for the reader. Table 4 now shows the data for ECT and RB together, allowing a comparison. |

[revised manuscript text omitted]

**5.5 [..[209]]Influence of the reference class definitions and the base rate**

So far we have explored ECT and RB assuming that there are no misclassifications of slope stability. However, as the true
415 slope stability is often not known (particularly in stable cases), errors in slope stability classification will occur. Such errors, however, may potentially influence all the statistics derived to describe the performance of tests (Brenner and Gefeller, 1997). For instance, if there are at least some slopes misclassified, classification performance will drop. However, in such cases, POD and PON will additionally be influenced by the true (though unknown) base rate (Brenner and Gefeller, 1997).
In previous studies exploring ECT (Moner et al., 2008; Simenhois and Birkeland, 2009; Winkler and Schweizer, 2009), slope
420 stability classifications were generally well described and the base rate for the applied slope stability classification given. However, slope stability classification approaches differed somewhat. For instance, a stability criterion used by Moner et al. (2008) was the occurrence of an avalanche on the test slope, while Simenhois and Birkeland (2009) additionally considered [..[210]]explosives testing of the slope as relevant information. Winkler and Schweizer (2009), on the other hand, additionally considered the manual profile classification used operationally in the Swiss avalanche warning service (Schweizer and Wiesinger,
425 2001; Schweizer, 2007)[..[211] ]. They already considered a location as *unstable*, when profiles were rated as «very poor» or «poor». As this classification relies rather strongly on the RB result, the RB would be favored in such an analysis (Winkler and Schweizer, 2009).
We have no knowledge about the uncertainty linked to our classification. However, we can demonstrate the impact of variations
* * *
[205]removed: and regardless of the strength of evidence,

[206]removed: varies

[207]removed: plane

[208]removed: plane

[209]removed: The influence

[210]removed: explosives-testing

[211]removed: and considered a sufficient criterion for instability

in the definition of the reference class on summary statistics like POD and PON, and using different data subsets for analysis:

430   Let us assume we are not interested in comparing ECT and RB, but want to explore only the performance of a binary ECT classification with [..²¹² ]*ECTP22* as the threshold between two classes. We will, however, use the RB together with the criteria introduced in Section 2.3 to define slope stability:

- Without using the RB as an additional [..²¹³ ]criterion, POD and PON for the ECT was [..²¹⁴ ]0.56 and 0.79, respectively (Fig. 4c).

435   - If only slopes [..²¹⁵ ]were considered *unstable*, when the RB stability class was $\leq 2$, and those as *stable* with RB stability class [..²¹⁶ ]4, the resulting POD [..²¹⁷ ]was 0.70 and PON [..²¹⁸ ]was 0.91. The base rate in this data set [..²¹⁹ ]was 0.32 and N = [..²²⁰ ]243.

- Being even more restrictive, and considering only slopes *unstable*, when the RB stability class was 1, and those as *stable* with RB stability class 4, the resulting POD [..²²¹ ]was 0.74 and PON was 0.91. The base rate in this data set [..²²² ]was 0.2 and N = [..²²³ ]206.

440

Of course, one could also be interested in exploring the performance of a binary classification of the RB, and define slope stability by using ECT results as additional [..²²⁴ ]criterion to those in Section 2.3. Without relying on ECT results, POD and PON for the RB were [..²²⁵ ]0.53 and 0.88, respectively (Fig. 4d). Considering only slopes as *unstable*, when additionally $ECT_{new}$ stability class $\leq 2$ was observed, and those with $ECT_{new}$ class 4 as [..²²⁶ ]*stable*, POD and PON would increase to

445   0.66 and [..²²⁷ ]0.94 (N = [..²²⁸ ]307, base rate 0.29), or 0.71 and [..²²⁹ ]0.94, respectively when considering only $ECT_{new}$ stability class 1 as *unstable* and class 4 as *stable* (N = [..²³⁰ ]285, base rate 0.23).

The combination of various error sources (Sect. 5.4), together with varying definitions of slope stability and differences in the base rate make it almost impossible to directly compare results obtained in different studies. Therefore, performance values
* * *
²¹²removed: ECTP22
²¹³removed: criteria
²¹⁴removed: 0.58 and 0.77
²¹⁵removed: are
²¹⁶removed: $\geq 3$
²¹⁷removed: is
²¹⁸removed: is 0.84
²¹⁹removed: is 0.14
²²⁰removed: 591.
²²¹removed: is 0.75 and PON is 0.89
²²²removed: is 0.14
²²³removed: 294.
²²⁴removed: criteria
²²⁵removed: 0.54 and 0.87
²²⁶removed: *unstable*, else as
²²⁷removed: 0.91
²²⁸removed: 561
²²⁹removed: 0.93
²³⁰removed: 385

[revised manuscript text omitted]